# Saliency-Based Rotor Spatial Position Displacement Self-Sensing for Self-Bearing Machines

**DOI:** 10.3390/s22249663

**Published:** 2022-12-09

**Authors:** Ye gu Kang, Daniel Fernandez, David Diaz Reigosa

**Affiliations:** 1School of Electrical, Electronics and Communication Engineering, Koreatech University, Chunan-si 31253, Republic of Korea; 2Electrical, Electronic, Computers and Systems Engineering, University of Oviedo, 33204 Oviedo, Spain

**Keywords:** bearingless permanent magnet synchronous machines (BPMSMs), high-frequency signal injection (HFI), rotor translational displacement, rotor eccentricity, spatial position, *xy*-position self-sensing

## Abstract

Self-bearing machines do not contain physical bearings but magnetic bearings. Both rotor rotary and spatial positions displacement are required in these types of machines to control the rotor position while it is levitating. Self-bearing machines often use external sensors for *x* (horizontal) and *y* (vertical) spatial position measurement, which will result in additional cost, volume, complexity, and number of parts susceptible to failure. To overcome these issues, this paper proposes a xy-position estimation self-sensing technique based on both main- and cross-inductance variation. The proposed method estimates *x* and *y* position based on inductive saliency between two sets of three-phase coils. The proposed idea is applied on a combined winding self-bearing machine which does not require additional suspension force winding. No additional search coil placement for xy-position estimation is required. Therefore, the proposed algorithm can result in a compact size self-bearing machine that does not require external sensors for xy-position measurement and suspension force winding.

## 1. Introduction

Conventional rotary machines only allow movement in one degree of freedom (DOF) typically in a rotation axis, i.e., $\theta_{r}$, shown in Figure 1. Self-bearing machines (bearingless machines), which do not have physical bearings, require controlling spatial xy-position of the rotor in Figure 1 (i.e., xy; *x*- and *y*-axes being horizontal and vertical axes) [1]. The remaining axes of the rotor could be controlled actively by a combination of more than two cascade-connected self-bearing machines, or passively by using reluctance forces [2,3,4].

Conventional self-bearing machines typically comprise two sets of independent windings in the stator for producing torque and suspension force, i.e., separated winding (a torque winding and a suspension winding). For compact size self-bearing machines, combined winding that uses the same coils to create both suspension forces and torque current have been developed [5,6].

Position sensors for closed-loop rotor spatial (xy) position control can be classified into: eddy current based [1,7,8], Hall effect based [9], and optical based sensors [10]. All these sensors result in additional cost, volume, complexity, and number of parts susceptible to failure [10,11]. In [12], model reference adaptive system (MRAS) is proposed to enable position self-sensing in a high speed region. In a zero-to-low speed region, high-frequency injection (HFI) has been demonstrated to be a viable option for xy-position estimation both in magnetic bearings [13,14,15,16] and self-bearing machines [17,18,19,20]. However, previously published HFI techniques applied to self-bearing machines are limited to: (i) In [17,18], separated winding machines utilizing suspension force windings mutual coupling inductances are proposed; (ii) In [19,20], separate search coils are installed to reduce input signal noise level by enabling differential mode for spatial rotor position estimation. In [20], search coil placement arrangement is investigated for minimizing the unwanted influence between suspension windings and pulse-width modulation (PWM) noise. It can be concluded that the previously proposed HFI techniques in the literature require additional windings or hardware which will result in increasing system volume and cost. Table 1 summarizes the existing HFI position self-sensing technique for self-bearing machines.

This paper proposes an HFI based xy-rotor position self-sensing technique for combined winding machines, therefore, achieving a compact size self-bearing system minimizing cost, volume, and reliability issues since it does not require additional winding or hardware. The combined winding machine has each phase connected to an inverter terminal that carries both suspension force and torque current. The target self-bearing machine winding is classified as multi-phase (MP), which has two sets of three-phase winding with two neutral points as in Figure 2 to enable dual field-oriented control for independent torque and force control [6,21]. The high-frequency (HF) signal will be injected in both sets of three-phase coils to enable use of differential mode input for xy-position estimation in a stationary reference frame. It will be shown that main- and cross-inductances variation depend on the rotor *x*- and *y*-positions, respectively, which will enable xy-position estimation self-sensing. The proposed HFI based self-sensing technique will be evaluated on a combined winding, 4-pole, 6-slot surface permanent magnet bearingless synchronous machine (SPMBSM).

The paper is organized as follows: HF voltage model of the target BPMSM for xy-position self-sensing is developed in Section 2. Section 3 introduces an analytical model that relates inductive saliency and rotor xy-position displacement. Section 4 presents the xy-position self-sensing method and its implementation. Section 5 presents experimental results; Finally, in Section 6, conclusions are drawn.

## 2. High-Frequency Model of a Self-Bearing Machine

The self-bearing machine voltage characteristic model is presented in this section. The inductive saliency model as a function of *x* and *y* displacement, later, will be used for the proposed self-sensing algorithm. The 4-pole, 6-slot combined winding self-bearing machine shown in Figure 1 will be used in this paper. Both the stator and the rotor core are soft magnetic material. The machine has two sets of three-phase windings ($a_{1}$, $b_{1}$, $c_{1}$, and $a_{2}$, $b_{2}$, $c_{2}$) with separated neutral points to control direction of flux induced from armature current. The two sets of three-phase windings are placed 180${}^{{}^{\circ}}$ mechanical angle apart from each other:(1)$$\begin{array}{cc} v_{an} & =R_{p}i_{an}+\frac{d\lambda_{an}}{dt}\\ v_{an} & =R_{p}i_{an}+L_{anan}\left(x,y\right)\frac{di_{an}}{dt}+L_{anbn}\left(x,y\right)\frac{di_{bn}}{dt}\\ & +L_{ancn}\left(x,y\right)\frac{di_{cn}}{dt}+L_{anam}\left(x,y\right)\frac{di_{am}}{dt}+L_{anbm}\left(x,y\right)\frac{di_{bm}}{dt}\\ & +L_{ancm}\left(x,y\right)\frac{di_{cm}}{dt}+k_{e\theta_{r}an}\left(x,y\right)\frac{d\theta_{r}}{dt}+k_{exan}\left(x,y\right)\frac{dx}{dt} \end{array}$$
(2)$$\begin{array}{c} +k_{eyan}\left(x,y\right)\frac{dy}{dt} \end{array}$$

### 2.1. Phase Voltage Model

Phase $a_{n}$ voltage characteristic model is shown in (Equation 1), where $R_{p}$, $v_{an}$, $i_{an}$, and $\lambda_{an}$ represent phase resistance, phase voltage, current, and flux linkage, respectively. *n* and *m* represent the first and the second sets of three-phase coils in Figure 2. Phase $a_{n}$ voltage characteristic equation can also be expressed by (Equation 2), where $L_{anan}$ is self-inductance, $L_{anbn}$ and $L_{ancn}$ represent mutual-inductances, $L_{anam}$, $L_{anbm}$, $L_{ancm}$, are cross-coupling inductance between coil sets *n* and *m*. $k_{e\theta ran}$, $k_{exan}$, and $k_{eyan}$ are back-EMF variables with respect to $\theta_{r}$-, *x*-, and *y*-position movement [1,9]. Note that the self- and mutual-inductances vary when the rotor spatial position (*x*, *y*) change. This is due to the unbalanced air-gap length between phases [22].

### 2.2. $\alpha \beta_{n}$-Axes Voltage Model

Six-phase voltage models can be presented as two sets of $\alpha \beta_{n}$-axes (Equation 4) and (Equation 5). When *n* is equal to 1, *m* is equal to 2, and vice versa. Clark transform, $K_{c}$, in (Equation 3) is used.
(3)$$\begin{array}{cc} \left[\begin{array}{c} \alpha_{n}\\ \beta_{n} \end{array}\right] & =\frac{2}{3}\left[\begin{array}{ccc} 1 & -0.5 & -0.5\\ 0 & \frac{\sqrt{3}}{2} & -\frac{\sqrt{3}}{2} \end{array}\right]\left[\begin{array}{c} a_{n}\\ b_{n}\\ c_{n} \end{array}\right]=K_{c}\left[\begin{array}{c} a_{n}\\ b_{n}\\ c_{n} \end{array}\right] \end{array}$$
(4)$$\begin{array}{cc} v_{\alpha n} & =R_{p}i_{\alpha n}+L_{\alpha n\alpha n}\left(x,y\right)\frac{di_{\alpha n}}{dt}+L_{\alpha n\beta n}\left(x,y\right)\frac{di_{\beta n}}{dt}\\ & +L_{\alpha n\alpha m}\left(x,y\right)\frac{di_{\alpha m}}{dt}+L_{\alpha n\beta m}\left(x,y\right)\frac{di_{\beta m}}{dt}\\ & +k_{e\theta_{r}\alpha n}\left(x,y\right)\frac{d\theta_{r}}{dt}+k_{ex\alpha n}\left(x,y\right)\frac{dx}{dt}+k_{ey\alpha n}\left(x,y\right)\frac{dy}{dt}\\ v_{\beta n} & =R_{p}i_{\beta n}+L_{\beta n\beta n}\left(x,y\right)\frac{di_{\beta n}}{dt}+L_{\beta n\alpha n}\left(x,y\right)\frac{di_{\alpha n}}{dt}\\ & +L_{\beta n\beta m}\left(x,y\right)\frac{di_{\beta m}}{dt}+L_{\beta n\alpha m}\left(x,y\right)\frac{di_{\alpha m}}{dt} \end{array}$$
(5)$$\begin{array}{c} +k_{e\theta_{r}\beta n}\left(x,y\right)\frac{d\theta_{r}}{dt}+k_{ex\beta n}\left(x,y\right)\frac{dx}{dt}+k_{ey\beta n}\left(x,y\right)\frac{dy}{dt} \end{array}$$

### 2.3. High-Frequency Voltage Model

The HF voltage characteristic model in stationary reference frame is represented by (Equation 6) and (Equation 7), obtained from (Equation 4) and (Equation 5) after neglecting the resistance terms, eliminating back-EMF terms, and cross-coupling inductances between three-phase coil sets. If the frequency of the HF signal feeding the machine is sufficiently high, the resistive terms can be safely neglected as the inductive terms dominate the machine impedance. The back-EMF term can also be eliminated as it does not contain HF component. The cross-coupling inductance between three-phase coil sets are small and insensitive to xy-position variation (see Figure 3).

It will be shown in Section 3 that $L_{\alpha n\alpha n}$, $L_{\beta n\beta n}$, $L_{\alpha n\beta n}$, and $L_{\beta n\alpha n}$ are a function of the *x*- and *y*-spatial rotor position displacement. This dependency will be used for rotor xy-position estimation: (6)$$\begin{array}{cc} v_{\alpha nHF} & =L_{\alpha n\alpha n}\left(x,y\right)\frac{di_{\alpha n}}{dt}+L_{\alpha n\beta n}\left(x,y\right)\frac{di_{\beta n}}{dt} \end{array}$$(7)$$\begin{array}{cc} v_{\beta nHF} & =L_{\beta n\beta n}\left(x,y\right)\frac{di_{\beta n}}{dt}+L_{\beta n\alpha n}\left(x,y\right)\frac{di_{\alpha n}}{dt} \end{array}$$

Finally, since $\alpha \beta_{2}$ is rotated 180${}^{\circ }$ with respect to $\alpha \beta_{1}$, the relationships shown in (Equation 8)–(Equation 11) exist among inductances in $\alpha \beta_{1}$ and $\alpha \beta_{2}$ reference frames.
(8)$$\begin{array}{cc} L_{\alpha 2\alpha 2}\left(x,y\right) & =L_{\alpha 1\alpha 1}\left(-x,-y\right) \end{array}$$
(9)$$\begin{array}{cc} L_{\alpha 2\beta 2}\left(x,y\right) & =L_{\alpha 1\beta 1}\left(-x,-y\right) \end{array}$$
(10)$$\begin{array}{cc} L_{\beta 2\beta 2}\left(x,y\right) & =L_{\beta 1\beta 1}\left(-x,-y\right) \end{array}$$
(11)$$\begin{array}{cc} L_{\beta 2\alpha 2}\left(x,y\right) & =L_{\beta 1\alpha 1}\left(-x,-y\right) \end{array}$$

## 3. HF Inductance Dependency and Rotor xy-Position Displacement

This section presents the inductance dependency in rotor spatial position displacement (i.e., xy-position). A simplified air-gap reluctance model is developed for the analysis of the self-bearing machine inductances sensitivity on rotor xy-position displacement. The conclusions drawn from the analysis are verified using finite element analysis (FEA).

### 3.1. Inductance Dependency Analysis Based on Air-Gap Reluctance Model

Equivalent inductance, $L_{eq}$, is shown in (Equation 12) as a function of reluctance, where $N_{eq}$ represents the number of turns. The reluctance in the air-gap is defined by (Equation 13), where $\mu_{0}$, $A_{eq}$, and $l_{eq}$ are the absolute permeability of the air, equivalent area, and air-gap length, respectively: (12)$$\begin{array}{cc} L_{eq} & =\frac{N_{eq}^{2}}{R} \end{array}$$(13)$$\begin{array}{cc} R & =\frac{l_{eq}}{\mu_{0}A_{eq}} \end{array}$$

Simplified magnetic equivalent circuit of the first set of three-phase coils, $abc_{1}$, is developed, assuming the air-gap reluctances are the dominant components in the flux-path. The reluctance components in the flux path of the rotor and stator are negligible and not included in the machine model. Infinite relative permeability of soft-iron is assumed. The relative permeability of PM is set equal to air. The three-phase magnetic circuit is shown in Figure 4, where $I_{a1}$, $I_{b1}$, $I_{c1}$, $\phi_{a1}$, $\phi_{b1}$, and $\phi_{c1}$ are $abc_{1}$ phase currents and $abc_{1}$ fluxes.

The $abc_{1}$ phase flux-linkage can be expressed by (Equation 21), solving Figure 4 using superposition [23], where phase flux-linkages $\lambda_{a1}$, $\lambda_{b1}$, and $\lambda_{c1}$ are obtained from multiplying the number of turns to the phase flux.

The phase flux-linkages in (Equation 21) can be expressed in an $\alpha \beta_{1}$ reference frame as in (Equation 22) applying Clark transformation. The HF inductances, $L_{\alpha 1\alpha 1}$, $L_{\alpha 1\beta 1}$, $L_{\beta 1\alpha 1}$, and $L_{\beta 1\beta 1}$ in (Equation 6) and (Equation 7) are derived using the phase air-gap reluctances. It is important to note that each phase reluctance is a function of the air-gap length (Equation 13). Therefore, if the rotor xy-position varies, the HF inductances will vary. The rotor should be designed with the least dq-axes inductive saliency to utilize the xy-position inductive saliency based on air-gap length variation for xy-position self-sensing.

### 3.2. $\alpha \beta_{n}$ Inductance Sensitivity Analysis

In this subsection, inductance sensitivity analysis is performed. Air-gap length in each phase is modeled as in (Equation 14) using Euler’s formula to correlate each phase air-gap length in polar coordinates to the xy-position variation in Cartesian coordinates. $\theta_{\phi }$ is the angle of the phase reluctance referencing the location of phase a1 as in Figure 4. *r* is the per-unit air-gap length. Δx and Δy are the per-unit rotor displacement in *x*- and *y*-axes. The effect of Δx and Δy on two sets of three-phase air-gap is modeled using the relative angle, $\theta_{\phi }$, in Table 2.
(14)$$\begin{array}{cc} l_{eq} & =|r-\left(\Delta x+j\Delta y\right)exp\left(j\theta_{\phi }\right)| \end{array}$$

Taking the dominant components, per unit air-gap length is calculated in (Equation 15)–(Equation 20): (15)$$\begin{array}{cc} l_{a1} & =\sqrt{1-2\Delta x} \end{array}$$(16)$$\begin{array}{cc} l_{b1} & =\sqrt{1+\Delta x+\sqrt{3}\Delta y} \end{array}$$(17)$$\begin{array}{cc} l_{c1} & =\sqrt{1+\Delta x-\sqrt{3}\Delta y} \end{array}$$(18)$$\begin{array}{cc} l_{a2} & =\sqrt{1+2\Delta x} \end{array}$$(19)$$\begin{array}{cc} l_{b2} & =\sqrt{1-\Delta x-\sqrt{3}\Delta y} \end{array}$$(20)$$\begin{array}{cc} l_{c2} & =\sqrt{1-\Delta x+\sqrt{3}\Delta y} \end{array}$$
(21)$$\begin{array}{cc} \left[\begin{array}{c} \lambda_{a1}\\ \lambda_{b1}\\ \lambda_{c1} \end{array}\right] & =\left[\begin{array}{c} N_{eq}\phi_{a1}\\ N_{eq}\phi_{b1}\\ N_{eq}\phi_{c1} \end{array}\right]\\ & =\frac{N_{eq}^{2}}{R_{a1}R_{b1}+R_{a1}R_{c1}+R_{b1}R_{c1}}\left[\begin{array}{ccc} \left(R_{b1}+R_{c1}\right) & -R_{c1} & -R_{b1}\\ -R_{c1} & \left(R_{a1}+R_{c1}\right) & -R_{a1}\\ -R_{b1} & -R_{a1} & \left(R_{a1}+R_{b1}\right) \end{array}\right]\left[\begin{array}{c} i_{a1}\\ i_{b1}\\ i_{c1} \end{array}\right]\\ \left[\begin{array}{c} \lambda_{\alpha 1}\\ \lambda_{\beta 1} \end{array}\right] & =\left[\begin{array}{cc} L_{\alpha 1\alpha 1} & L_{\alpha 1\beta 1}\\ L_{\beta 1\alpha 1} & L_{\beta 1\beta 1} \end{array}\right]\left[\begin{array}{c} i_{\alpha 1}\\ i_{\beta 1} \end{array}\right] \end{array}$$
(22)$$\begin{array}{c} =\frac{N_{eq}^{2}}{2(R_{a1}R_{b1}+R_{a1}R_{c1}+R_{b1}R_{c1})}\left[\begin{array}{cc} 3(R_{b1}+R_{c1}) & \sqrt{3}\left(R_{b1}-R_{c1}\right)\\ \sqrt{3}\left(R_{b1}-R_{c1}\right) & 4R_{a1}+R_{b1}+R_{c1} \end{array}\right]\left[\begin{array}{c} i_{\alpha 1}\\ i_{\beta 1} \end{array}\right] \end{array}$$

Equivalent air-gap area, $A_{eq}$, and air-gap permeability are assumed equal for each phase. Substituting the phase reluctance to (Equation 22), taking Taylor’s series expansion near center position, (Equation 23) is calculated as function of *x* and *y* rotor position displacement, Δx and Δy. Similarly, the second set of inductances is calculated in (Equation 24). The sensitivity analysis based air-gap reluctance is verified using FEA in Figure 5 and Figure 6 in Section 3.2: (23)$$\begin{array}{cc} \left[\begin{array}{cc} L_{\alpha 1\alpha 1} & L_{\alpha 1\beta 1}\\ L_{\beta 1\alpha 1} & L_{\beta 1\beta 1} \end{array}\right] & =\frac{N_{eq}^{2}A_{eq}\mu_{0}}{2*3}\left[\begin{array}{cc} 6+3\Delta x & 3\Delta y\\ 3\Delta y & 6-3\Delta x \end{array}\right] \end{array}$$(24)$$\begin{array}{cc} \left[\begin{array}{cc} L_{\alpha 2\alpha 2} & L_{\alpha 2\beta 2}\\ L_{\beta 2\alpha 2} & L_{\beta 2\beta 2} \end{array}\right] & =\frac{N_{eq}^{2}A_{eq}\mu_{0}}{2*3}\left[\begin{array}{cc} 6-3\Delta x & -3\Delta y\\ -3\Delta y & 6+3\Delta x \end{array}\right] \end{array}$$

The conclusions of inductance variation in *x* and *y* rotor position displacement from the simplified inductance model can be summarized as follows:Main inductances, $L_{\alpha n\alpha n}$ and $L_{\beta n\beta n}$, are sensitive to *x*-axis rotor movement. For *x*-position self-sensing, main inductances are required.Cross-coupling inductances, $L_{\alpha n\beta n}$ and $L_{\beta n\alpha n}$, are sensitive to *y*-axis rotor movement. For *y*-position self-sensing, cross-coupling inductances are required.Cross-coupling inductances, $L_{\alpha n\beta n}$ and $L_{\beta n\alpha n}$, are equal.The main and cross-coupling inductances in the first and the second set of three-phase coils have opposite sensitivity to xy-axes rotor movement.

### 3.3. Inductance Dependency Verification Using FEA

Two-dimensional static finite element analysis (FEA) is performed to verify the conclusions from the simplified reluctance model. A stacking factor of 0.95 is used for the stator with N35 grade NdFeB permanent magnet on the rotor. Self-bearing machine geometry and simulation conditions for FEA analysis are summarized in Table 3, where $D_{os}$, $D_{is}$, $D_{or}$, $w_{t}$, $t_{PM}$, and $L_{st}$ are stator outer and inner diameter, outer diameter of the rotor, teeth width, thickness of the magnet, and stack length; see Figure 7. 0.2 p.u. of HF current is injected in the first set of three-phase coils, *n*, while the other set of three-phase coils current, *m*, are set to zero (Equation 25) and (Equation 26). Inductances are calculated based on FEA by taking partial derivative of HF flux-linkage with respect to the HF current (Equation 27)–(Equation 30). Given *n* = 1, *m* = 2, *i* = 1, *j* = 1, main and cross-coupling inductances shown in Figure 5 are calculated for the first set of coil inductances while *n* = 2, *m* = 1, *i* = 1, *j* = 1 calculates the second set of the coil inductance in Figure 6. The trends of inductance variation are matching with analysis based on air-gap reluctances in Section 3.2.

The cross-coupling inductance between the first and the second set of three-phase coils, shown in Figure 3, are calculated using *n* = 1, *m* = 2, *i* = 1, *j* = 2. Note that the magnitude of the cross-coupling inductances between three-phase coil sets are much smaller (about 1/5 times) than the self inductances shown in Figure 5 and Figure 6. Moreover, the inductances are not sensitive to xy-position variation. This is because the air-gap reluctance sensitivity of abc1 and abc2 to xy-position variation are in opposite directions (see Table 2 with angle offset, $\theta_{\phi }$, of π between abc1 and abc2). This is due to the air-gap reluctance sensitivity of αβ1 and αβ2 are in π shifted (opposite) relations (see Table 2). Therefore, the coupling inductances between three-phase coil sets are not included in the xy-position self-sensing model.
(25)$$\begin{array}{cc} \left[\begin{array}{c} i_{\alpha n}\\ i_{\beta n} \end{array}\right] & =K_{c}\left[\begin{array}{c} i_{an}\\ i_{bn}\\ i_{cn} \end{array}\right] \end{array}$$
(26)$$\begin{array}{cc} \left[\begin{array}{c} i_{\alpha m}\\ i_{\beta m} \end{array}\right] & =\left[\begin{array}{c} 0\\ 0 \end{array}\right] \end{array}$$
(27)$$\begin{array}{cc} L_{\alpha i\alpha j}\left(x,y\right) & =\frac{\partial \lambda_{\alpha i}\left(x,y\right)}{\partial i_{\alpha j}} \end{array}$$
(28)$$\begin{array}{cc} L_{\alpha i\beta j}\left(x,y\right) & =\frac{\partial \lambda_{\alpha i}\left(x,y\right)}{\partial i_{\beta j}} \end{array}$$
(29)$$\begin{array}{cc} L_{\beta i\beta j}\left(x,y\right) & =\frac{\partial \lambda_{\beta i}\left(x,y\right)}{\partial i_{\beta j}} \end{array}$$
(30)$$\begin{array}{cc} L_{\beta i\alpha j}\left(x,y\right) & =\frac{\partial \lambda_{\beta i}\left(x,y\right)}{\partial i_{\alpha j}} \end{array}$$

## 4. HFI Based xy-Position Self-Sensing

xy-position estimation based on the injection of an HF signal is presented.

Equations (Equation 6) and (Equation 7) can be expressed as (Equation 31), where *s* is the differential operator. Park transform in (Equation 32) with reference frame offset angle $\theta_{inj}$ is defined to rotate the stationary reference frame. Pulsating HF voltages are injected in between αβ-axes (Equation 33) where $\theta_{inj}$ of 45º is chosen to excite both α- and β-axes simultaneously (as shown in Figure 8) [24]. The same magnitude HF voltage command is used for both sets of three-phase coil set (*n* = 1, *n* = 2) as in (Equation 33) to use differential information to enhance position estimation accuracy.
(31)$$\begin{array}{cc} \left[\begin{array}{c} v_{\alpha nHF}\\ v_{\beta nHF} \end{array}\right] & =s\left[\begin{array}{cc} L_{\alpha n\alpha n} & L_{\alpha n\beta n}\\ L_{\beta n\alpha n} & L_{\beta n\beta n} \end{array}\right]\left[\begin{array}{c} i_{\alpha nHF}\\ i_{\beta nHF} \end{array}\right] \end{array}$$
(32)$$\begin{array}{cc} K_{p}\left(\theta_{inj}\right) & =\left[\begin{array}{cc} cos(\theta_{inj}) & sin(\theta_{inj})\\ -sin(\theta_{inj}) & cos(\theta_{inj}) \end{array}\right] \end{array}$$
(33)$$\begin{array}{cc} \left[\begin{array}{c} v_{\alpha nHF}^{\theta_{inj}}\\ v_{\beta nHF}^{\theta_{inj}} \end{array}\right] & =V_{HF}\cos\left(2\pi f_{HF}t\right)\left[\begin{array}{c} 1\\ 0 \end{array}\right] \end{array}$$
(34)$$\begin{array}{cc} \left[\begin{array}{c} v_{\alpha nHF}^{*}\\ v_{\beta nHF}^{*} \end{array}\right] & =K_{p}{\left(45\right)}^{-1}\left[\begin{array}{c} v_{\alpha nHF}^{\theta_{inj}}\\ v_{\beta nHF}^{\theta_{inj}} \end{array}\right]\\ \left[\begin{array}{c} i_{\alpha nHF}^{\theta_{inj}}\\ i_{\beta nHF}^{\theta_{inj}} \end{array}\right] & =K_{p}\left(45\right)\left[\begin{array}{c} i_{\alpha nHF}\\ i_{\beta nHF} \end{array}\right] \end{array}$$
(35)$$\begin{array}{c} =\sin\left(2\pi f_{HF}t\right)\left[\begin{array}{c} I_{i0n}\\ I_{i1n} \end{array}\right] \end{array}$$
(36)$$\begin{array}{cc} \left[\begin{array}{c} I_{i0n}\\ I_{i1n} \end{array}\right]\; & =\frac{V_{HF}}{2\pi f_{HF}\left(L_{\alpha n\alpha n}L_{\beta n\beta n}-L_{\alpha n\beta n}L_{\beta n\alpha n}\right)}\left[\begin{array}{c} \frac{\left(L_{\alpha n\alpha n}+L_{\beta n\beta n}\right)}{2}-\frac{\left(L_{\alpha n\beta n}+L_{\beta n\alpha n}\right)}{2}\\ -\frac{\left(L_{\alpha n\alpha n}-L_{\beta n\beta n}\right)}{2}-\frac{\left(L_{\alpha n\beta n}-L_{\beta n\alpha n}\right)}{2} \end{array}\right] \end{array}$$

$V_{HF}$ and $f_{HF}$ represent the HF voltage magnitude and frequency. $f_{HF}$ should be selected far apart from the controlled torque or force current frequency to minimize the interference. The HF voltage command input in the stationary reference frame will be (Equation 34). The resulting stator HF current response will be Park transformed to the HF injection reference frame as in (Equation 35), where $I_{i0n}$ and $I_{i1n}$ are injection reference frame HF current magnitude defined in (Equation 36).

This question arises: How can this HF current response can be used to estimate *x* and *y* rotor position? This may be explained referring to (Equation 23), (Equation 24), and (Equation 36). The main inductances, $L_{\alpha n\alpha n}$ and $L_{\beta n\beta n}$, are proportional to Δx. On the other hand, the cross-coupling inductances, $L_{\alpha n\beta n}$ and $L_{\beta n\alpha n}$, are sensitive to Δy. The inductance term in the denominator of (Equation 36), $\frac{V_{HF}}{2\pi f_{HF}}\frac{1}{\left(L_{\alpha n\alpha n}L_{\beta n\beta n}-L_{\alpha n\beta n}L_{\beta n\alpha n}\right)}$ becomes $\frac{V_{HF}}{2\pi f_{HF}}\frac{36}{\left(N_{eq}A_{eq}\mu_{0}\right)\left(36-9{\left(\Delta x\right)}^{2}-9{\left(\Delta y\right)}^{2}\right)}$ since the term is insensitive to small Δx and Δy. $N_{eq}$, $A_{eq}$, $\mu_{0}$, $V_{HF}$, and $f_{HF}$ are assumed to be constants. Therefore, $I_{i0n}$ and $I_{i1n}$ have the xy-position dependency relationship shown in (Equation 37) and (Equation 38).

Exciting both $\alpha \beta_{1}$ and $\alpha \beta_{2}$ with (Equation 33) as in Figure 8, *x* and *y* rotor position are estimated, from the HF current responses, (Equation 39) and (Equation 40). Figure 9 shows FEA based results of differential HF currents in (Equation 39) and (Equation 40). The definitions of $k_{gx}$ and $k_{ox}$ are shown in Figure 9a, where $k_{gx}$ is *x*-axis proportional gain for unit conversion from HF current, i.e., amperes (A), to position, e.g., millimeter (mm), given HF injection conditions, i.e., $V_{HF}$, $f_{HF}$. $k_{ox}$ is the *x*-axis offset constant to calibrate the *x*-axis referencing position. The procedure to select the self-sensing gains is as follows:first, $k_{gx}$ and $k_{gy}$ are set to 1 while searching for the values of $k_{ox}$ and $k_{oy}$ that will result in $\hat{x}$ = $\hat{y}$ = 0 in (Equation 39) and (Equation 40) at the rotor position *x* = *y* = 0 mm. $k_{ox}$ and $k_{oy}$ are non-zero values if the two sets of three-phase are not in perfect symmetry.second, $k_{gx}$ and $k_{gy}$ are selected to convert the estimated position unit to the unit in millimeters.
(37)$$\begin{array}{cc} \left[\begin{array}{c} I_{i01}\\ I_{i11} \end{array}\right]\; & \approx \left[\begin{array}{c} \frac{\left(L_{\alpha 1\alpha 1}+L_{\beta 1\beta 1}\right)}{2}-\frac{\left(L_{\alpha 1\beta 1}+L_{\beta 1\alpha 1}\right)}{2}\\ -\frac{\left(L_{\alpha 1\alpha 1}-L_{\beta 1\beta 1}\right)}{2}-\frac{\left(L_{\alpha 1\beta 1}-L_{\beta 1\alpha 1}\right)}{2} \end{array}\right]\\ \; & \propto \left[\begin{array}{c} \frac{\left(\Delta x-\Delta x\right)}{2}-\frac{\left(\Delta y+\Delta y\right)}{2}\\ -\frac{\left(\Delta x+\Delta x\right)}{2}-\frac{\left(\Delta y-\Delta y\right)}{2} \end{array}\right]=\left[\begin{array}{c} -\Delta y\\ -\Delta x \end{array}\right] \end{array}$$
(38)$$\begin{array}{cc} \left[\begin{array}{c} I_{i02}\\ I_{i12} \end{array}\right]\; & \approx \left[\begin{array}{c} \frac{\left(L_{\alpha 2\alpha 2}+L_{\beta 2\beta 2}\right)}{2}-\frac{\left(L_{\alpha 2\beta 2}-L_{\beta 2\alpha 2}\right)}{2}\\ -\frac{\left(L_{\alpha 2\alpha 2}-L_{\beta 2\beta 2}\right)}{2}-\frac{\left(L_{\alpha 2\beta 2}-L_{\beta 2\alpha 2}\right)}{2} \end{array}\right]\\ \; & \propto \left[\begin{array}{c} \frac{\left(-\Delta x+\Delta x\right)}{2}-\frac{\left(-\Delta y-\Delta y\right)}{2}\\ -\frac{\left(-\Delta x-\Delta x\right)}{2}-\frac{\left(-\Delta y+\Delta y\right)}{2} \end{array}\right]=\left[\begin{array}{c} \Delta y\\ \Delta x \end{array}\right] \end{array}$$
(39)$$\begin{array}{cc} \hat{x} & =k_{gx}\left(\left(I_{i12}-I_{i11}\right)+k_{ox}\right) \end{array}$$
(40)$$\begin{array}{cc} \hat{y} & =k_{gy}\left(\left(I_{i02}-I_{i01}\right)+k_{oy}\right) \end{array}$$

**Figure 9 sensors-22-09663-f009:**
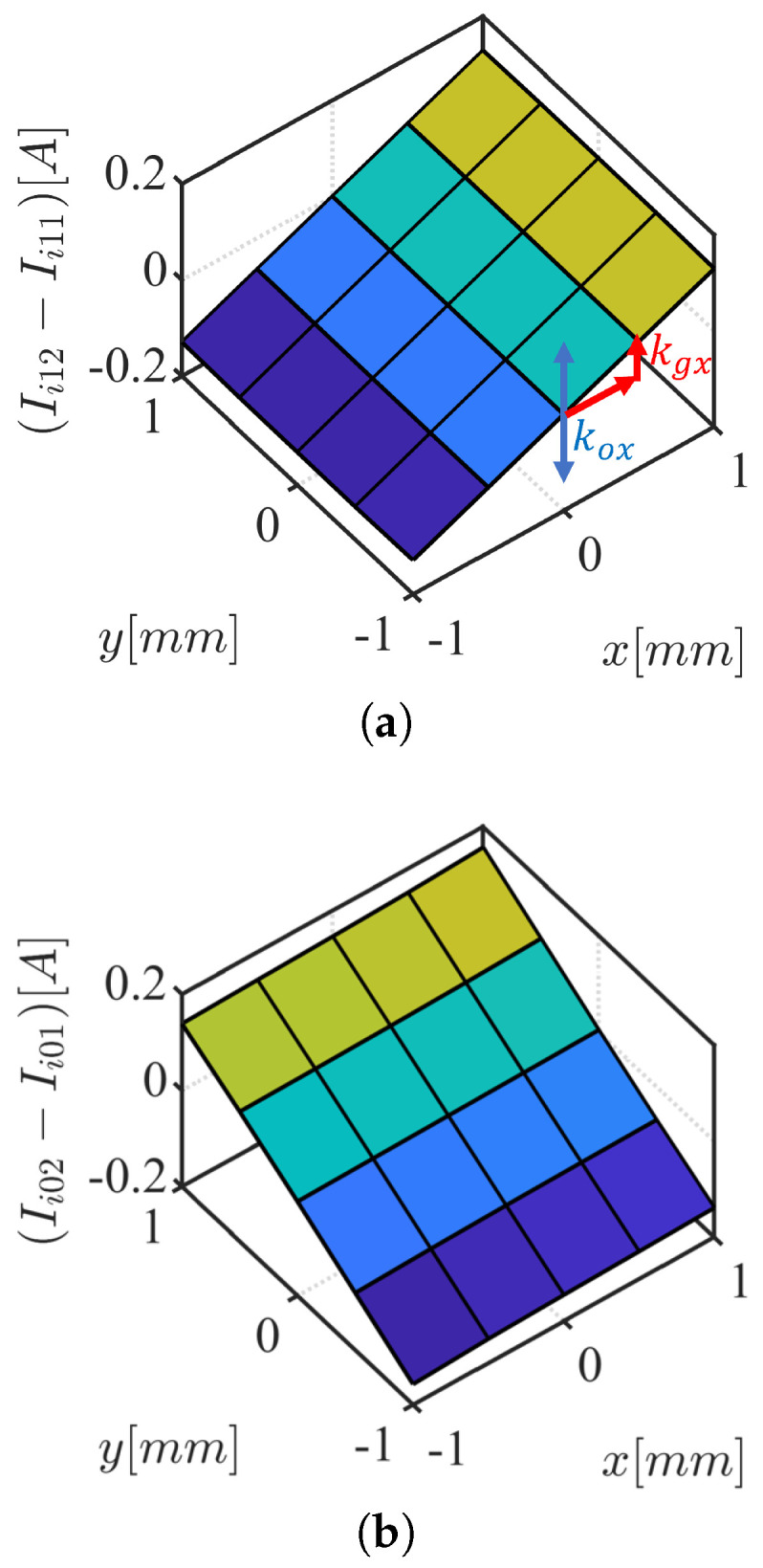
FEA results of differential HF currents, $V_{HF}$ = 1 V, $f_{HF}$ = 1000 Hz, and gain and offset constant, $k_{gx}$ and $k_{ox}$. (**a**) ($I_{i12}-I_{i11}$); (**b**) ($I_{i02}-I_{i01}$).

### xy-Position Self-Sensing Implementation

Figure 10 presents the overall xy-position self-sensing block diagram. An HF pulsating voltage is injected in 45${}^{{}^{\circ}}$ injection reference frame, i.e., in between $\alpha \beta_{1}$ and $\alpha \beta_{2}$ as shown in Figure 8. The commanded HF voltages are input into a self-bearing machine in (Equation 6) and (Equation 7) using a PWM voltage source inverter (VSI). Two sets of three-phase current responses are measured using 6-current transducers and Clark transformed. HF current responses, $I_{i0n}$ and $I_{i1n}$ containing xy-position information, are rotated back to $\alpha \beta_{1}$ and $\alpha \beta_{2}$ reference frame. The HF current is demodulated with a low pass filter (LPF) after multiplying HF current with sine wave at injected frequency, $f_{HF}$. Finally, *x* and *y* rotor positions are estimated using (Equation 39) and (Equation 40).

## 5. Experimental Results

This section shows the experimental results of the xy-position self-sensing algorithm. The experimental setup shown in Figure 11 is used. Figure 11a is the test self-bearing machine. To drive the test machine, two sets of 3-phase inverters with 6-current transducers are used, as shown in Figure 11b. The rotor is mechanically coupled with a DC machine that is fixed to the base, as shown in Figure 11b. The self-bearing machine stator is fixed to the xyz-mover of the position controlled linear motion stage. The linear motion stage shown in Figure 11c is position controlled by switched reluctance (SR) motors and drivers. The controlled position precision for *x*- and *y*-axes is 0.1 mm. The center position of the rotor with respect to the stator is position controlled by the linear motion stage, and used as an absolute position reference during the xy-position self-sensing experimentation.

The self-sensing experiment conditions are summarized in Table 4. Inverter DC link voltage is set to 40V. HF voltage ($v_{\alpha 1HF}^{*}$, $v_{\beta 1HF}^{*}$, $v_{\alpha 2HF}^{*}$, and $v_{\beta 2HF}^{*}$) shown in Figure 12a are injected. The resulting HF currents, when *x* = 0 mm and *y*= −1 mm, are shown in Figure 12b. Taking Park transformation at +45${}^{{}^{\circ}}$, HF currents are measured. $I_{i01}$, $I_{i11}$, $I_{i02}$, and $I_{i12}$ are shown in Figure 13a, which are obtained following demodulation procedure (green) shown in Figure 10. The pulsating HF current will result in pulsating HF torque and force that will not impact the average torque and force. Nevertheless, the HF injection condition, i.e., $f_{HF}$ and $V_{HF}$, should be chosen carefully to minimize the interference between HF injection induced torque and force and the controlled torque and force. The $f_{HF}$ is chosen as one-tenth of the switching frequency to minimize the impact of the switching voltage of PWM. The $V_{HF}$ of <3% of the DC link voltage of the inverter can be selected for the proposed self-sensing algorithm. The proposed algorithm is tested in open-loop assuming that the voltage excitation results in minimal impact to the drive system in closed-loop. Finally, Figure 13b illustrates the estimated *x* and *y* positions, $\hat{x}$ and $\hat{y}$, using (Equation 39) and (Equation 40). Using 500 Hz bandwidth LPF, the estimated position converges to the steady state value in 2 ms. Steady state position estimation error was 4%, and peak to peak estimation error was ±8% for maximum operating range of ±1 mm. The peak to peak error is due to a voltage synthesize error from PWM VSI. The position estimation bandwidth, which affects position feedback control, is limited to LPF bandwidth. To achieve high bandwidth for the self-sensing, a higher PWM switching frequency should be used.

To further show viability of the xy-position self-sensing, a series of experimental results on $\hat{x}$, $\hat{y}$ are shown in Figure 14 and Figure 15. In Figure 14, the center of the rotor is held at constant *y*-positions while the *x*-positions are varied from −1 mm to 1 mm in incremental steps of 0.5 mm, see Figure 11c. Experimental conditions are summarized in Table 4. xy-position self-sensing block diagram shown in Figure 10 is used. Steady state value of demodulated HF currents and estimated $\hat{x}$-, $\hat{y}$-position are shown in Figure 14. It can be observed that $I_{i11}$ and $I_{i12}$ in Figure 14c,d are reverse-sensitive to the *x*-position variation while $I_{i0n}$ in Figure 14a,b is insensitive. Using (Equation 39) and (Equation 40), $\hat{x}$ and $\hat{y}$ positions are estimated, simultaneously, and shown in Figure 14e,f. Figure 14g,h shows the estimation error.

Figure 15 shows analogous results, but this time, the *y*-position is varied while holding the *x*-position constant. $I_{i0n}$ is sensitive to the *y* position as expected in (Equation 37) and (Equation 38). The estimated *x*- and *y*-position results are shown in Figure 15e,f, respectively. Figure 15g,h shows the estimation error. The position estimation error increases as the rotor moves away from the center position due to other machine nonlinearity, e.g., magnetic saturation effect.

## 6. Conclusions

Rotor xy-position estimation based on the injection of an HF signal in combined winding self-bearing machines has been proposed. The proposed xy-position self-sensing algorithm does not require additional windings, e.g., suspension winding or search coils.

The model for machine inductances’ sensitivities with respect to the xy-rotor position displacement are developed, based on the air-gap reluctance model, verified by FEA and experimental results. An HF pulsating voltage is injected between α- and β-axes to enable estimation of *x*- and *y*-rotor position, simultaneously utilizing both main- and cross-coupling inductance variation. The excitation frequency is carefully selected to minimize the interference to the closed-loop operation of the bearingless machine drive system. Demodulating the coupled HF current, $\hat{x}$ and $\hat{y}$ rotor positions are estimated. Experimental results have been provided to support the viability of the proposed technique.

## Figures and Tables

**Figure 1 sensors-22-09663-f001:**
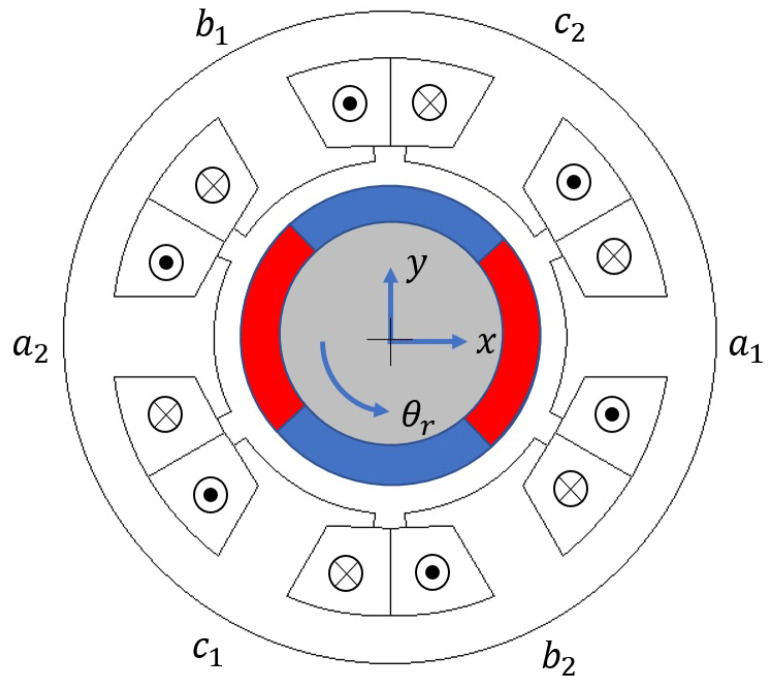
Four-pole, six-slot, combined winding self-bearing machine.

**Figure 2 sensors-22-09663-f002:**
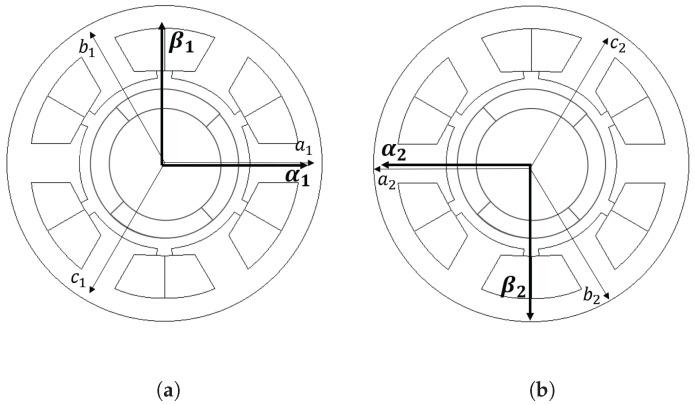
abc to αβ transformations: (**a**) $a_{1}$, $b_{1}$, $c_{1}$, and $\alpha_{1}$, $\beta_{1}$; (**b**) $a_{2}$, $b_{2}$, $c_{2}$, and $\alpha_{2}$, $\beta_{2}$.

**Figure 3 sensors-22-09663-f003:**
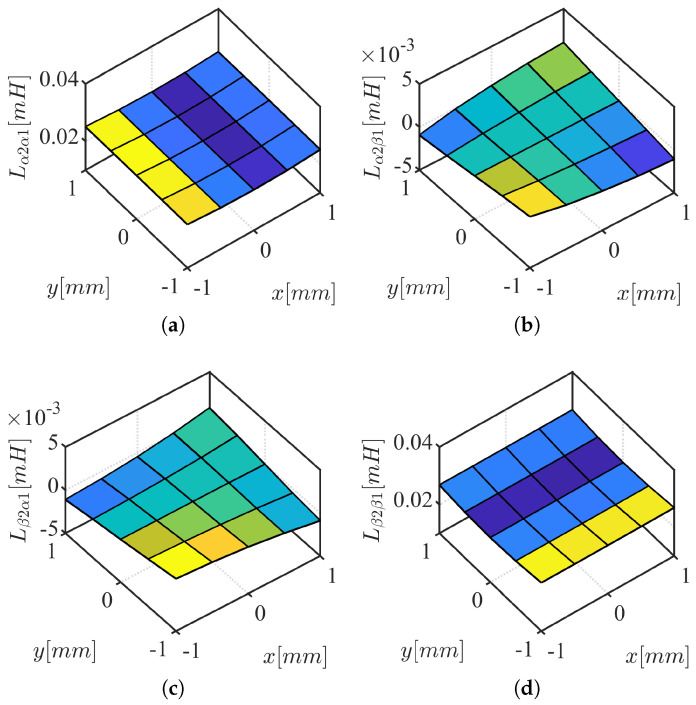
$\alpha \beta_{2}$ three-phase cross-coupling inductance results from FEA: (**a**) $L_{\alpha 2\alpha 1}$; (**b**) $L_{\alpha 2\beta 1}$; (**c**) $L_{\beta 2\alpha 1}$; (**d**) $L_{\beta 2\beta 1}$.

**Figure 4 sensors-22-09663-f004:**
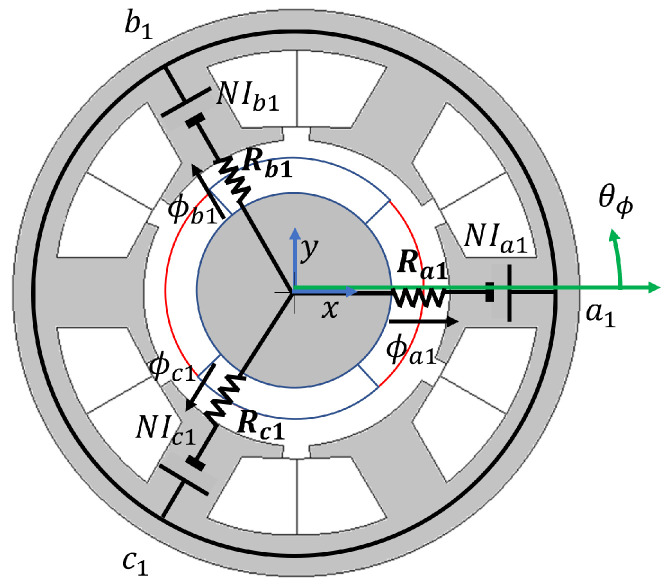
Simplified magnetic equivalent circuit of $abc_{1}$ based on air-gap reluctance.

**Figure 5 sensors-22-09663-f005:**
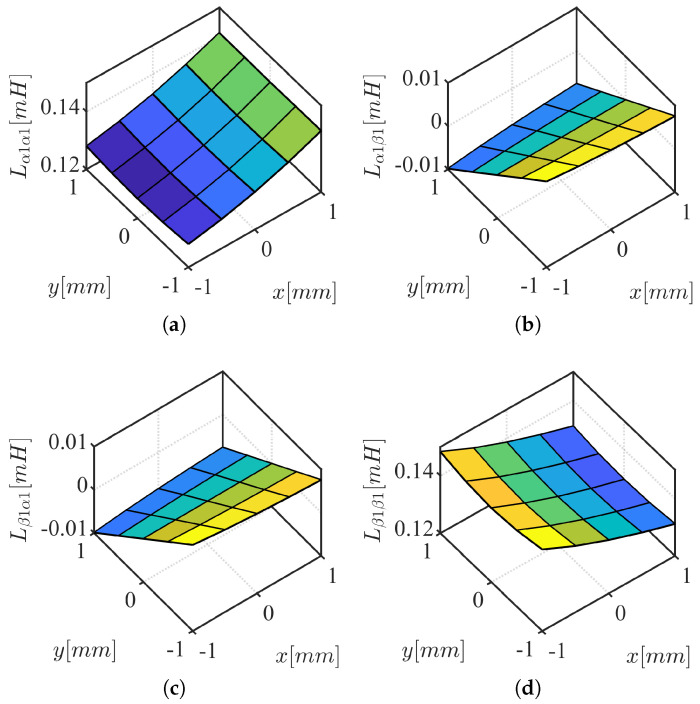
$\alpha \beta_{1}$ inductance results from FEA: (**a**) $L_{\alpha 1\alpha 1}$; (**b**) $L_{\alpha 1\beta 1}$; (**c**) $L_{\beta 1\alpha 1}$; (**d**) $L_{\beta 1\beta 1}$.

**Figure 6 sensors-22-09663-f006:**
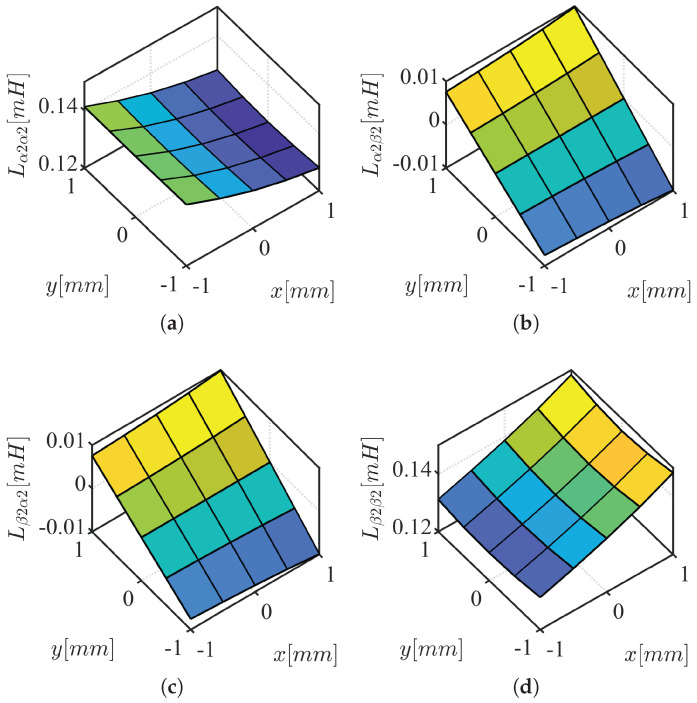
$\alpha \beta_{2}$ inductance results from FEA: (**a**) $L_{\alpha 2\alpha 2}$; (**b**) $L_{\alpha 2\beta 2}$; (**c**) $L_{\beta 2\alpha 2}$; (**d**) $L_{\beta 2\beta 2}$.

**Figure 7 sensors-22-09663-f007:**
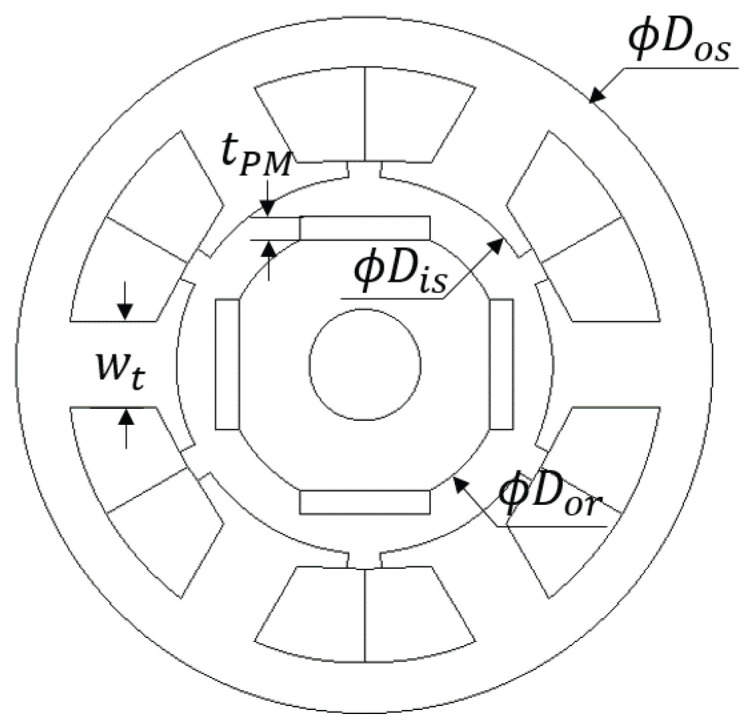
Cross-section of target SPMBSM and relevant geometric variables for FEA.

**Figure 8 sensors-22-09663-f008:**
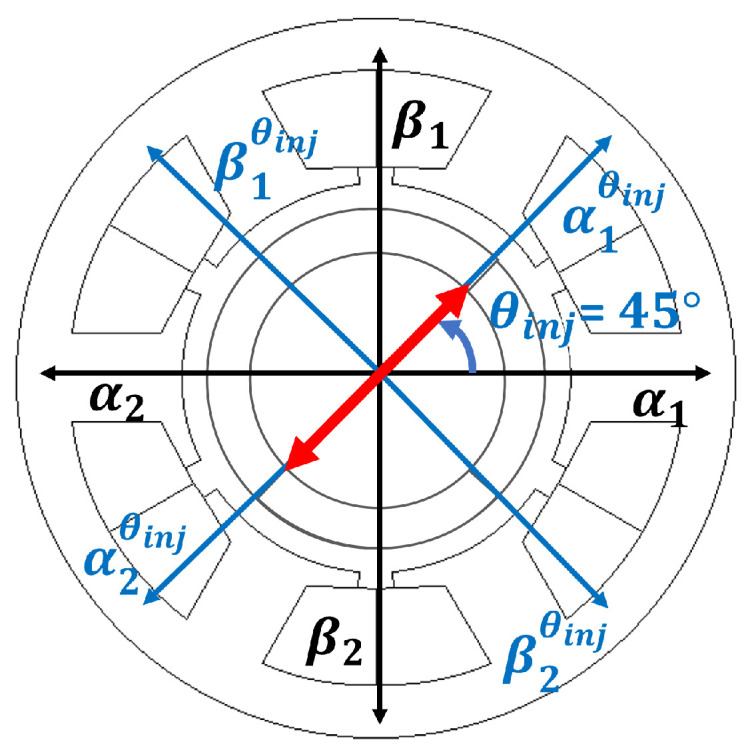
HF pulsating voltage injection reference frame, $\alpha \beta_{1}^{\theta_{inj}}$ and $\alpha \beta_{2}^{\theta_{inj}}$ with $\theta_{inj}$, and stationary reference frame, $\alpha \beta_{1}$ and $\alpha \beta_{2}$.

**Figure 10 sensors-22-09663-f010:**
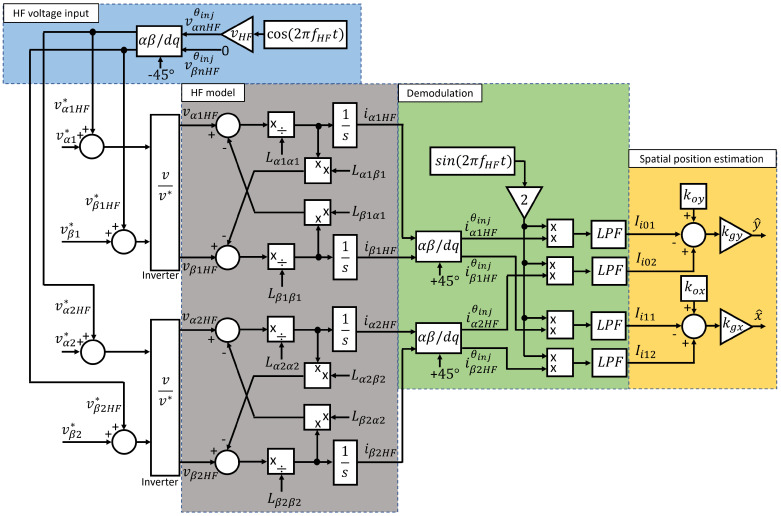
xy-position self-sensing block diagram, HF voltage injection in between αβ-axes (blue), HF model of the self-bearing machine (gray), HF current demodulation block diagram (green), and xy-position estimation block diagram (orange).

**Figure 11 sensors-22-09663-f011:**
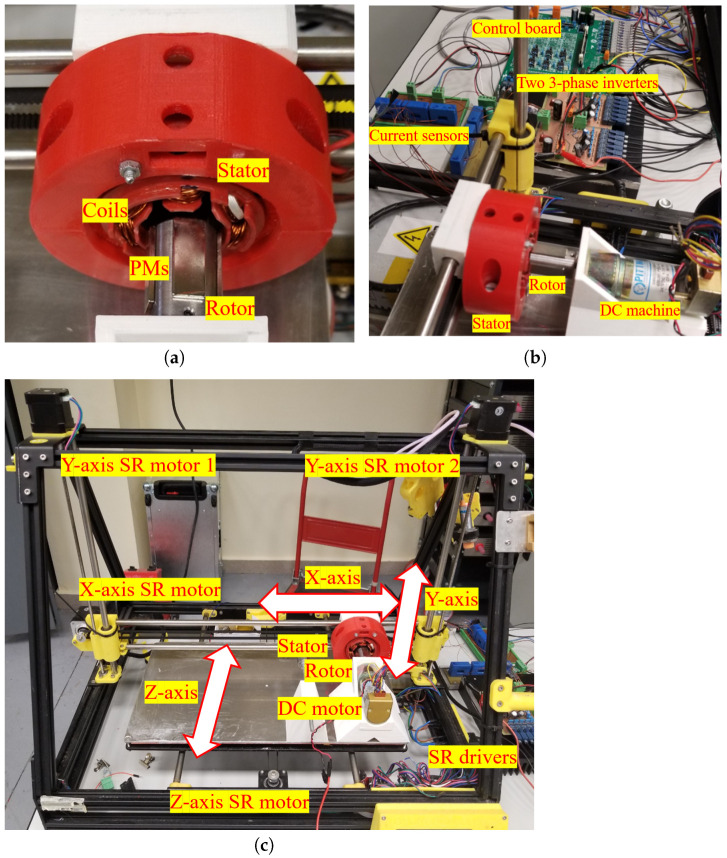
Experimental setup. (**a**) self-bearing machine; (**b**) self-bearing machine driver; (**c**) xyz linear motion stage.

**Figure 12 sensors-22-09663-f012:**
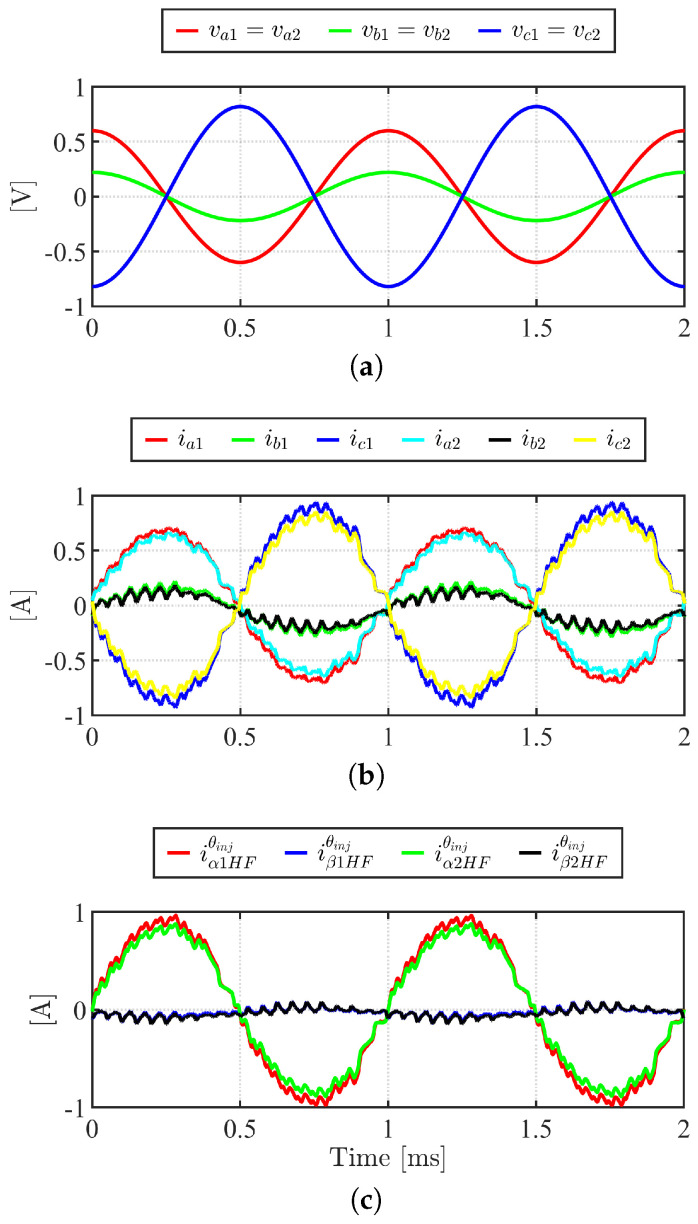
HF pulsating voltage injection and current response, *x* = 0 mm, *y* = −1 mm. (**a**) HF phase voltage command, $abc_{1}$ and $abc_{2}$; (**b**) HF phase current response, $abc_{1}$ and $abc_{2}$; (**c**) HF current response of $i_{\alpha 1HF}^{\theta_{inj}}$, $i_{\beta 1HF}^{\theta_{inj}}$, $i_{\alpha 2HF}^{\theta_{inj}}$, and $i_{\beta 2HF}^{\theta_{inj}}$.

**Figure 13 sensors-22-09663-f013:**
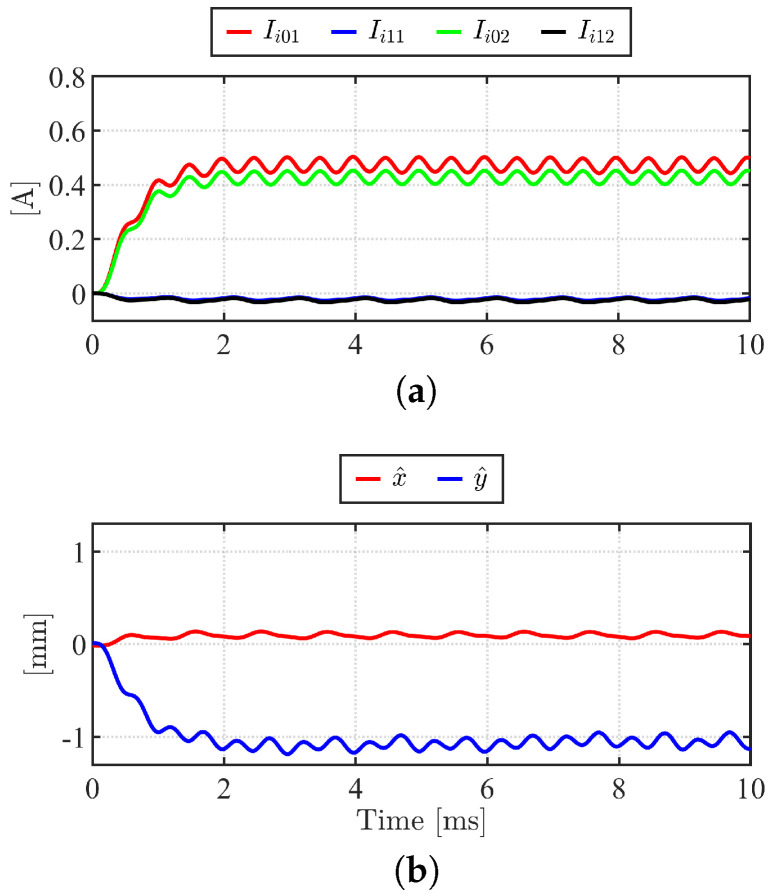
xy-position estimation experimental results, *x* = 0 mm, *y* = −1 mm. (**a**) demodulated α and β HF current, $I_{i01}$, $I_{i11}$, $I_{i02}$, and $I_{i11}$; (**b**) estimated xy-position, $\hat{x}$ and $\hat{y}$.

**Figure 14 sensors-22-09663-f014:**
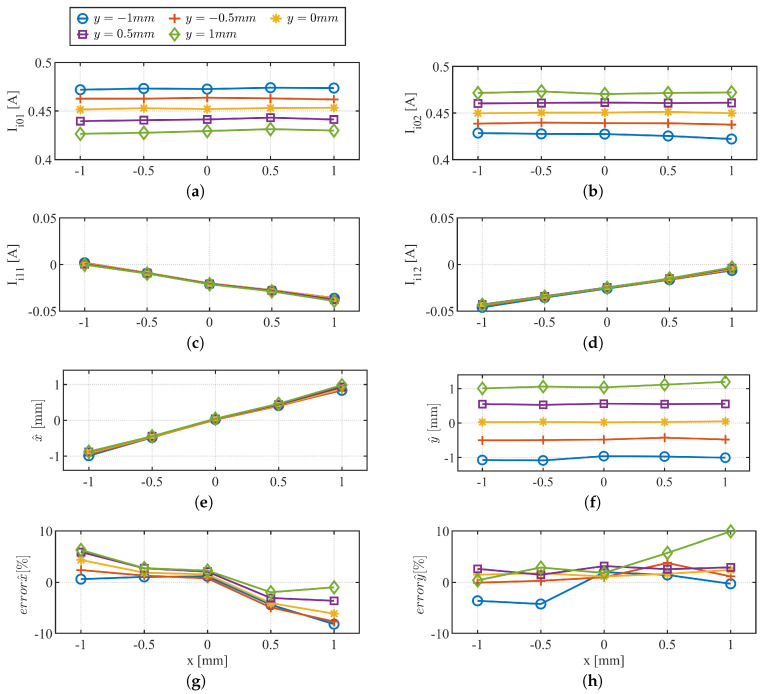
Experimental results varying *x* from −1 mm to +1 mm in steps of 0.5 mm at constant *y* position. (**a**) $I_{i01}$; (**b**) $I_{i02}$; (**c**) $I_{i11}$; (**d**) $I_{i12}$; (**e**) $\hat{x}$; (**f**) $\hat{y}$; (**g**) $error\hat{x}$; (**h**) $error\hat{y}$.

**Figure 15 sensors-22-09663-f015:**
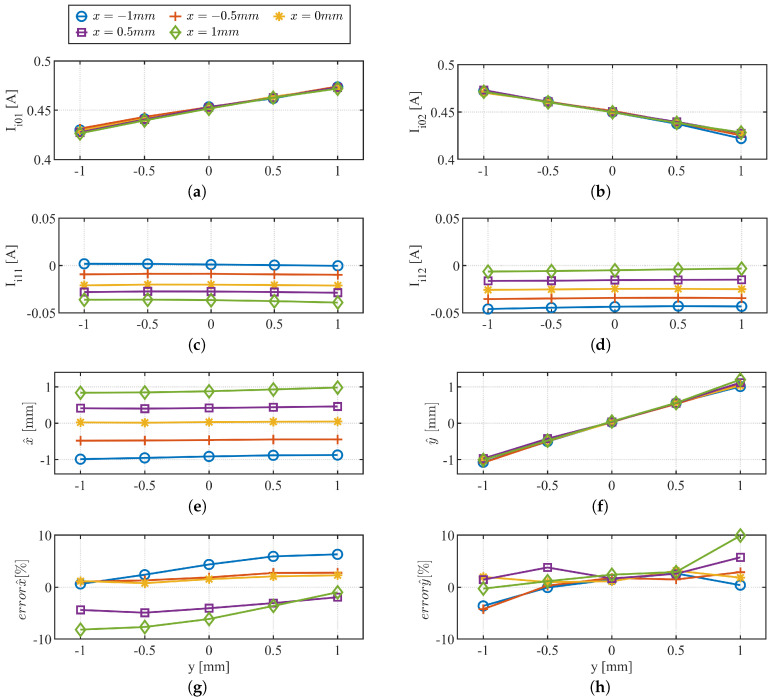
Experimental results varying *y* from +1 mm to −1 mm in steps of 0.5 mm at constant *x* position. (**a**) $I_{i01}$; (**b**) $I_{i02}$; (**c**) $I_{i11}$; (**d**) $I_{i12}$; (**e**) $\hat{x}$; (**f**) $\hat{y}$; (**g**) $error\hat{x}$; (**h**) $error\hat{y}$.

**Table 1 sensors-22-09663-t001:** HFI spatial position self-sensing methods comparison.

	Methodology		
	Input Signal Mode	Require Additional Winding	Winding Type
[17,18]	Common	No	Separated
[19,20]	Differential	Yes ${}^{1}$	Separated
Proposed	Differential	No	Combined

^1^ Additional hardware placement is required.

**Table 2 sensors-22-09663-t002:** Phase equivalent air-gap length model.

Air-Gap Length	$\theta_{\phi }$ [rad]	Air-Gap Length	$\theta_{\phi }$ [rad]
$l_{a1}$	0	$l_{a2}$	π
$l_{b1}$	2π/3	$l_{b2}$	π+2π/3
$l_{c1}$	−2π/3	$l_{c2}$	π−2π/3

**Table 3 sensors-22-09663-t003:** FE analysis conditions and self-bearing machine.

Poles	4	$D_{os}$	50 [mm]
Slots	6	$D_{is}$	26.8 [mm]
Turns	16	$D_{or}$	19.6 [mm]
Permanent magnet	N35	$w_{t}$	6 [mm]
Soft material	M-19	$t_{PM}$	1.4 [mm]
$L_{st}$	24.5 [mm]	$I_{rated}$	5 [A]
$x_{max}$	±2 [mm]	*x*	−1 to 1 in 0.5 [mm] step
$y_{max}$	±2 [mm]	*y*	−1 to 1 in 0.5 [mm] step

**Table 4 sensors-22-09663-t004:** xy self-bearing experimental conditions.

$V_{DC}$	40 [V]	$k_{gx}$	23.5
$V_{HF}$	0.015 [p.u.]	$k_{ox}$	0.0058
$f_{HF}$	1000 [Hz]	$k_{gy}$	23.5
LPF bandwidth	500 [Hz]	$k_{oy}$	−0.0006

## Data Availability

Not applicable.

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
