# Peer review of "Saliency-Based Rotor Spatial Position Displacement Self-Sensing for Self-Bearing Machines"

_sensors, 2022, doi:10.3390/s22249663_

Round 1
Reviewer 1 Report
The paper presents a very interesting spatial position estimation technique which is intended to be used with bearingless PM synchronous machine.
The estimation method is based on the machine model so the robustness of the estimation technique towards parameters variation should be evaluated.
Is there any condition required about the machine saliency limits for the developed method to be effective?
As authors state, fHF and VHF should be chosen carefully to minimize the interference between HF injection and the fundamental frequency voltage that produce the necessary torque. However, there is not enough details about this issue in the paper.
The impact of the PWM on the estimation procedure and results should be explained as well.
I suggest you to add the position estimation error graph for both x and y axis.
The paper presents stationary tests only, and the Fig. 13 shows 1ms estimation dynamic for fixed x, and y positions. I suggest you to add experimental results concerning complete test with rotating motor or at least explain how mature is the presented technique to be integrated in sensorless bearingless synchronous machine Drives.
Finally, the conclusion should be improved.
Author Response
Please, see the attached PDF file.
Thank you for your help.

Reviewer 2 Report
This paper presents a xy-position estimation self-sensing technique based on both main- and cross-inductance variation. In my opinion, the content of this article is interesting, but the authors should address the following issues:
1. The existing literature shows that a built neural network can estimate faster and better. In fact, the full text is the idea of soft measurement. Now the popular one is neural network estimation. Why does the author still use traditional methods without considering advanced means such as neural network.
2. How the estimated performance is verified by the author when the load changes. That is to say, more experimental conditions need to be tested to verify the performance.
Author Response
Please, see the attachment.
Thank you.

Round 2
Reviewer 2 Report
None